# Resting vagal tone, alpha amylase and cortisol levels in women with eating disorders before and after psychotherapy

Simone Munsch [1‡*], Nadine Messerli-Bürgy[2‡], Marius Rubo[3], Andrea H. Meyer[1,4], Kathrin Schopf[5], Nadine Humbel[1], Felicitas Forrer[1], Dirk Adolph[5], Jürgen Margraf[5], Silvia Schneider[5]

1 Clinical Psychology and Psychotherapy, Department of Psychology, Food Research and Innovation Center, FRIC, University of Fribourg, Fribourg, Switzerland, 2 Clinical Child and Adolescent Psychology, Department of Psychology, University of Lausanne, Lausanne, Switzerland, 3 Cognitive Psychology, Perception and Research Methods, Institute of Psychology, University of Bern, Bern, Switzerland, 4 Department of Psychology, Division of Clinical Psychology and Epidemiology, University of Basel, Basel, Switzerland, 5 Faculty of Psychology, Mental Health Research and Treatment Center, Ruhr University Bochum, Bochum, Germany

‡ These authors share first authorship on this work.
* simone.munsch@unifr.ch

## Abstract

### Background

Eating disorders such as Anorexia Nervosa (AN) and Bulimia Nervosa (BN) were previously found to partly entail alterations in stress physiology including salivary cortisol (sC), and salivary alpha amylase (sAA) at rest and basal vagal tone (HF-HRV), compared to individuals without mental disorders or with mixed mental disorders (anxiety and depressive disorders), but corresponding data remain scarce and are not entirely consistent.

### Method

HF-HRV, sC and sAA at rest were assessed in a female sample of 58 individuals with AN and 54 individuals with BN before and after psychotherapy and contrasted against measurements from 59 female individuals suffering from mixed disorders and 101female healthy controls.

### Results

Values for sC were elevated in AN compared to all other groups, those for HF-HRV were highest in both AN and BN and lowest in mixed mental disorders and no differences were found at rest for sAA. During psychotherapy, HF-HRV changed more in AN and BN groups than in HC or mixed samples. sC and sAA remained unchanged. There was no association between BMI and stress physiology.

**Data availability statement:** Data cannot be shared publicly because at the time of assessment, open data access was not a standard procedure, and participants were not informed that their anonymized data would be shared with other researchers without restrictions. Therefore, anonymised data are available from the Departmental Institutional Ethical Review Board, IRB, University of Fribourg, Prof. Petra Vetter (petra.vetter@unifr.ch).

**Funding:** The present research was supported by grants from several funding agencies (SNSF 149416 and 170503, DFG SCHN 415/4, SANS 22 12, Research Fund University of Fribourg: 419), but the funding agencies were not involved in our study design, data collection and analysis, decision to publish, or preparation of the manuscript.

**Competing interests:** The authors have declared that no competing interests exist.

## Conclusion

Alterations in stress physiology present differently across EDs and mixed mental disorders. Correlates of physiological functioning remained mostly stable throughout 3 months of psychotherapy. Only basal vagal tone was normalized in AN/BN in comparison to HC. This might indicate that physiological changes can occur early, but mostly take longer to change during treatment.

**Trial registration:** Data were assessed during a multi-site cross- and longitudinal experimental trial registered at the German Clinical Trials Registry (trial number: DRKS00005709; see [1] for details).

## 1. Introduction

Eating disorders (ED) entail alterations in stress physiology, which may contribute to the onset and maintenance of ED symptoms [1]. An individual's physiological capacity to adapt to current environmental conditions represents an important mechanism for coping with daily environmental challenges [2]. This capacity depends on the flexible neural network system called the autonomic nervous system (ANS) which is connected to the amygdala and the medial prefrontal cortex and represents the most important influence on the variability of the heart rate (HRV) [3] and therefore one branch of the stress regulation system. HRV is more than a proxy for the healthy functioning of the heart; it is also an index of the integrative capacity of the brain to adapt to complex environmental demands flexibly. Based on the Model of Neurovisceral Integration [4], high-frequency heart rate variability (HF-HRV) representing parasympathetic activation (PNS) (vagal tone) is a biological marker of emotion regulation capacity. Difficulties in regulating emotions represent a transdiagnostic core feature. They are a treatment target within the onset period and the course of a broad range of mental disorders and ED [5]. Higher HF-HRV in resting state and greater HF-HRV changes when facing situational challenges are related to more adaptive emotion regulation attempts and subsequent recovery from a stressor.

Stress physiology and ED-related behavior are intertwined. For example, malnutrition and a history of extreme weight loss might contribute to a physiological dysregulation of the hypothalamic-pituitary-adrenal (HPA) axis (measured by cortisol levels (sC)) that could be reversible after weight restoration [6]. Such increased sC levels at rest have been found in AN or extreme weight conditions [7–9], but not in BN [10], while results in other mental disorders such as anxiety or depression remain contradictory [11–13]. ED-related behaviors such as self-starving, food restrictions, but also repeated binge eating, and compensatory behavior have been related to dysregulated stress physiology in previous studies, where overactivity of the vagal tone (representing autonomic nervous system (ANS) activity) was found in AN and BN [14–20]. This vagal overactivity in ED can lead to a destabilization of the vagal balance, an increase of the vagal threshold, and heightened basal vagal tone according to the desensitization hypothesis and, in consequence, to limited flexibility to respond to emotional stimuli [17]. These findings are partly supported by research

on another ANS biomarker, the salivary alpha amylase (sAA) which is enhanced at rest in BN [10] but not consistently in AN [10,12,21]. However, vagal overactivity seems to be an ED-specific condition as it has not been found in other mental disorders. Non-ED psychopathologies such as anxiety and depressive disorders (mixed mental disorders) or even healthy controls with abnormal weight rather show reduced vagal activity at rest representing chronic stress conditions [22–32], while findings on sAA at rest are inconsistent [33–35].

Substantial alleviation of ED symptoms after evidence-based psychotherapy could be expected to improve dysregulated stress physiology as psychological functioning and the physiological stress system tightly interact [4,36]. Previous studies revealed that enhanced reactivity of the autonomic nervous system normalized in patients with AN and BN after inpatient cognitive behavioral therapy (CBT) [37] and in those with BN after 16 sessions of inpatient CBT combined with medication [20]. By contrast, reduced autonomic activity at pre-treatment in individuals suffering from non-ED disorders did not improve during an inpatient nine-week psychotherapy [28] nor during an eight-week mindfulness-based cognitive outpatient therapy [38]. Also, according to meta-analytic evidence, the corresponding results on vagal tone for anxiety disorders are mixed [23], but improvements in dysregulated stress physiology might be found in other biomarkers, such as sC levels, where findings on changes after psychotherapy in ED are still limited. Only one study investigated sC reactivity but did not report sC at rest. They found that blunted cortisol reactivity in ED persisted through psychotherapy [37], but the influence of Body Mass Index (BMI) changes during treatment, a primary treatment target in AN [39], has not been investigated yet [9]. Further, results in non-ED pathology have been inconsistent and depended on psychotherapy characteristics (e.g., in- and outpatient psychotherapy, different duration, and number of sessions [40].

To sum up, even though there is a consensus regarding the importance of a better understanding of the physiological underpinnings of symptoms of ED, the current evidence on physiological mechanisms in EDs and non-ED (anxiety and depression) and their variability before and after treatment remains scarce.

Therefore, the aims of the present study were a) to compare stress physiology before treatment between individuals with ED and non-ED disorders including AN, BN, mixed group (depression and anxiety), and further healthy controls; and b) to investigate potential changes of stress physiology during evidence-based treatment options in different clinics.

More precisely, at pretreatment, **we first** expected elevated sC mean values in AN at pretreatment compared to the remaining groups of BN, mixed, and healthy controls (before waiting), elevated sAA mean values in BN and mixed compared to AN and healthy controls, and highest basal vagal tone values in the ED groups (AN and BN) with intermediate values for healthy controls and lowest values in the mixed group. For this hypothesis, we included the following four covariates: physical activity, smoking habit, medication intake with effects on the HPA axis, and use of contraceptives. Disregarding healthy controls, we further expected elevated sC mean values in AN compared to the clinical groups of BN and mixed, elevated sAA mean values in BN and mixed compared to AN, and higher basal vagal tone values in the ED groups (AN and BN) than in the mixed group. For this hypothesis, we included the same covariates as before, plus the following two covariates, which could not be included in the first hypothesis since they were completely confounded with the healthy control group: patient type (in-/outpatient), and comorbidity.

**Second**, we expected comparable changes during treatment (or during waiting for healthy controls) in stress physiology in clinical groups compared to healthy controls during waiting for sC and sAA, but a stronger decrease in AN and BN, and not in the mixed group, compared to the healthy control group for basal vagal tone. For this hypothesis, physical activity, smoking habit, medication intake with effects on the HPA axis, and use of contraceptives were included as covariates. Disregarding healthy controls, we further expected a higher decrease in sC, sAA and for basal vagal tone in AN and BN than in the mixed group, controlling for in- and outpatient setting, number of sessions and comorbidity. For this hypothesis, the 4 covariates, plus patient type (in-/outpatient), number of therapy sessions, and comorbidity, were included as covariates.

**Third**, at pretreatment, we expected for sC a negative correlation with BMI in AN and a positive correlation with BMI in the combined BN and mixed groups. For sAA levels, we expected a negative correlation with BMI in the combined clinical

groups, and for basal vagal tone, we expected a positive correlation with BMI in AN and a negative correlation with BMI in the combined BN and mixed groups. For this hypothesis, physical activity, smoking habit, medication intake with effects on the HPA axis, and use of contraceptives, plus patient type (in-/outpatient) and comorbidity, were included as covariates. **Finally**, fourth, we expected a temporal association between BMI and sC, sAA, and basal vagal tone during treatment in the AN group, including physical activity, smoking habit, medication intake with effects on the HPA axis, and use of contraceptives, plus patient type (in-/outpatient), number of therapy sessions and comorbidity as covariates.

## 2. Method

### 2.1. Participants and design

Data were assessed during a multi-site cross- and longitudinal experimental trial registered at the German Clinical Trials Registry (trial number: DRKS00005709; see [41] for details) to investigate the effects of cognitive distortion on media exposure before and after treatment [42]. Individuals with ED or mixed mental disorders were recruited at Swiss and German treatment sites and healthy controls at the University of Fribourg [42]. After the diagnostic procedure, altogether 213 Swiss and German females in- and outpatients were enrolled, including 130 women with ED (n = 64 with AN, n = 66 with BN), 83 women with mixed (depressive disorders, n = 54; anxiety disorders, n = 26; somatoform disorders, n = 3) and 128 healthy controls (HC). A total of 66 women did not meet inclusion criteria or withdrew from study participation (i.e., declined to participate, discontinued experiment or unavailable) (see, e.g., [42]). The final sample at pretreatment consisted of 272 young women with an age range of 18–35 years (M = 22.85, SD = 3.96); whereof 112 with ED (58 AN, 54 BN), 59 with mixed, and 101 healthy controls. Physiological data was defined as a secondary outcome in the overall trial, which explains the smaller sample size here than the estimated size of N = 309 as indicated in the published study protocol [41]. All participants received financial compensation or credit points (students) and were debriefed about the purpose of the study after completion. A total of 27 persons dropped out after treatment (AN n = 6, BN n = 7, mixed n = 13 and healthy n = 1), resulting in 245 participants (52 AN, 47 BN, 46 mixed and 100 healthy controls) at posttreatment.

During the recruitment period from 01/01/2014 until 10/08/2017, participants were included if female, aged between 18 and 35 years, meeting diagnostic criteria for an AN or BN (ED sample), and anxiety, depressive or somatoform disorder (mixed mental disorders sample), based on a structured clinical interview DIPS (Diagnostisches Interview für psychische Störungen [43]) and providing written informed consent. Exclusion criteria were current pregnancy, breastfeeding, or intake of beta-blockers. Women with mixed disorders were further excluded if they displayed clinically relevant symptoms of ED or a clinical diagnosis of ED [44]. Healthy participants were excluded if they showed any current mental disorder or past ED assessed by the DIPS or a body dysmorphic disorder assessed by the SCID (Structured Clinical Interview for DSM-IV Axis I, Section G, Body Dysmorphic Disorder [45]), or an EDE-Q total score (Eating Disorder Examination Questionnaire [46]) higher than 2.5 according to Fischer et al. (2012). The study protocol was approved by the Ethics Committees of the Canton of Fribourg and several other cantons of Psychology, Switzerland (reference no. 2012_001) and of the Faculty of Psychology at the Ruhr University Bochum in Germany (reference no. 142) and followed the guidelines of the Declaration of Helsinki and the Good Clinical Practice Directives of Switzerland.

### 2.2. Procedure

All individuals completed a diagnostic interview within the first week to assess their eligibility and participated in a laboratory experiment where psychological and physiological mechanisms during thin ideal exposure were measured [41,42]. These experiments were conducted in the afternoon (2 pm-4.30 pm). To reduce the impact of influencing factors, participants were told to avoid eating and drinking other than water, brush their teeth or use mouth rinse, and not smoke within one hour before the experiment. All participants were randomly assigned to one of two exposure conditions (thin ideal or neutral) and were told that the study examines mental well-being and psychophysiological stress reactivity relating to body

image satisfaction in young adults. Before and after this exposure condition, saliva was repeatedly sampled at a total of 9 different time points throughout the experiment including baseline periods, exposure conditions, and during recovery. Previous analyses revealed no change during these periods of the experimental task [42]. Therefore, for the analyses in this study, only the preparatory baseline period was considered, but neither exposure conditions nor recovery periods.

After this first testing afternoon, women of the healthy control group waited 3 months until the experiment was re-conducted, whereas women with mental disorders were treated as usual, following the national clinical practice treatment guideline for disorder-specific and were then reinvited to the experiment in their corresponding clinics after 3-months of treatment. The number of treatment sessions and treatment settings varied, with an average number of treatment sessions of 28.2 (AN), 13.6 (BN), and 15.3 (mixed). Regarding the partitioning into in- and outpatients, 32 (55%), 21 (39%), and 45 (76%) were inpatients and 26 (45%), 33 (61%), and 14 (24%) were outpatients in the AN, BN, and mixed group, respectively.

For women with AN, inpatient treatment was associated with large improvements in BMI ($d=.78$), depressive symptoms ($d=.77$), and moderate improvements in ED pathology ($d=.48$), while outpatient treatment was associated with small to medium improvements in BMI ($d=.37$). For women with BN, inpatient treatment was associated with large improvements in ED pathology ($d=.81$) and moderate improvements in depressive symptoms ($d=.70$), while outpatient treatment was associated with small to medium improvements in ED pathology ($d=.44$) and depressive symptoms ($d=.39$). For a more detailed description of the ED treatment and the treatment effects in terms of recovery and remission rates as well as symptomatic changes, we refer to [47]. For individuals in the mixed group, outpatient treatment was associated with small to medium improvements in ED pathology ($d=.40$), while inpatient treatment was associated with moderate improvements in depressive symptoms ($d=.60$).

### 2.3. Measures

**2.3.1. Diagnostic interview.** The DIPS (Diagnostisches Interview für psychische Störungen [48]) is a structured interview based on the DSM-IV-TR (Diagnostic and Statistical Manual of Mental Disorders [49]) with interrater reliability values ranging from.57 to.92, and retest-reliability values (Cohen's Kappa) ranging from.35 to.94 [43]. For our study, only the DIPS eating disorders section was adapted to the DSM-5 [43,50]. Interviewers were trained and supervised by the principal investigator. Ten percent of the interviews were coded twice by two independent raters based on the audio recordings of the interviews. Interrater reliability for primary diagnoses included the raters' and the interviewer's ratings and achieved high values (Fleiss K = .850; Fleiss K = .803).

**2.3.2. Weight and height.** Weight and height were assessed using an electronic personal scale (Seca 899, Basel, Switzerland) and a stadiometer (Seca, Basel, Switzerland) and BMI was calculated afterwards. Participants were asked to take their shoes off for these measurements. Use of medication influencing physiological stress branches were assessed.

**2.3.3. Vagal tone.** Basal vagal tone was assessed during a 2-minute resting period by using movisens ekgMove chest belts (ambulatory monitoring system; movisens GmbH, Karlsruhe, Germany) with a sampling frequency of 1024 Hz. Interbeat interval (R-R) data was visually inspected and spectral values (high-frequency band (HF-HRV)) were calculated based on the autoregressive method using Kubios software (University of Eastern Finland, Kuopio, Finland), considering a smooth priors detrending method ($\lambda=500$) and correction for artifacts. HRV-HF power values and normalized units (n.u.) were used for data analysis. Due to severe artifacts in some cases, only data of 220 participants could be included in the analyses.

**2.3.4. Salivary cortisol and salivary alpha amylase.** Saliva sample at rest was taken at the beginning and end of the resting period (5 minutes after arrival at the laboratory and again at 35 minutes) using Salivettes (Sarstedt, Nümbrecht, Germany). Saliva samples were frozen at –20°C at the study sites before sending off to the Laboratory of University of Zürich (Swiss samples) and the Laboratory of the Department of Genetic Psychology at Ruhr-University Bochum (German samples) for analyses. sC levels were analyzed on a Synergy2 plate reader (Biotek, USA), using

commercial enzyme-linked immunosorbent assays (ELISAs; free cortisol in saliva; Demeditec, Germany) according to the manufacturer's instructions. Intraassay and interassay coefficients of variation were < 9%. Sensitivity of the assay was 0.019 ng/ml. To measure sAA, a colorimetric test using 2-chloro-4-nitrophenyl-$\alpha$-maltrotriosoide (CNP-G3) as a substrate reagent was applied [51]. Intra-assay and inter-assay coefficients of variation were < 10%. Sensitivity was at.65 U/ml.

**2.3.5. Assessment of covariates.** Self-reported physical activity (hours per week), smoking habit (number of cigarettes per day), medication intake with effects on the HPA axis, and use of contraceptives was assessed through a questionnaire. Further, information on patient type (inpatient/outpatient/healthy), number of psychotherapy sessions, and comorbidity were collected and used as covariates to control for the heterogeneity of treatment intensity [47].

## 2.4. Data processing

A few data points were unavailable due to measurement errors (for sC data at pretreatment (n = 1 BN) and at posttreatment (n = 1 AN, n = 1 healthy control); for sAA at posttreatment (n = 1 with AN). Individual data points for HF-HRV were unavailable in 31 participants at the premeasurement (3 AN, 4 BN, 8 mixed, 16 healthy) and in 25 participants at the post measurement (3 AN, 3 BN, 3 Mixed, 16 healthy controls) due to measurement errors. All physiological parameters (HF-HRV, sC and sAA) were ln-transformed to meet the criteria for homoscedasticity and normality.

## 2.5. Data analysis

Data were analyzed using R (version 4.3.2). To compare physiological parameters (HF-HRV, sC, sAA) between specific groups before treatment (hypothesis 1), we used linear models, including the respective a priori between-group contrast, as specified in our hypotheses, and all covariates. To analyse the change in physiological parameters between pre- and posttreatment between the clinical group (sC, sAA) or the combined AN/BN groups (HF-HRV) and HC (hypothesis 2), we used linear mixed-effects models with time (pre-post treatment) as within-subjects factor and the respective combination of groups as between-subjects factor, including all covariates. For associations at pretreatment between BMI and physiological parameters for specific subgroups (hypothesis 3), we used partial correlation, thereby partialling out all covariates. Finally, to assess the association between BMI and physiological parameters over time (hypothesis 4), we used a multilevel model with subject-centered BMI as a time-varying predictor, including all covariates. The following covariates were included for all analyses: physical activity, smoking habit, medication intake with effects on the HPA axis, and use of contraceptives. In addition, comorbidity (y/n) and patient type (out- or inpatient treatment/healthy) were considered for hypotheses focusing on comparisons exclusively between the clinical groups. Further, we considered the number of therapy sessions as a covariate whenever changes during treatment were at the center of our analyses including only clinical groups as in parts of hypotheses 2 and 4. Results concerning covariates are only reported if noteworthy, i.e., if the p-value of the association with the respective outcome was smaller than.01. Effect sizes are reported for all results. For hypothesis 4 we used semi-partial $R^2_{\beta*}$ according to [52] to indicate effect sizes regarding associations with BMI. For family-wise correction of p-values (padj) we used the method by [53].

Power analyses ($\alpha$ = .05, $\beta$ = 0.2, one-sided test) were performed for all hypotheses. Hypothesis 1 that deals with a priori contrasts, refers to a two-group comparison, controlling for covariates. Assuming a medium effect size, the required sample size is c. 128. Hypothesis 2 deals with a two-factorial design with one-between (groups) and one within-subjects (time points) factor. Assuming a medium effect size and a serial correlation of.5, the required sample size is 34. Hypothesis 3 deals with partial correlation. For a medium effect size of r = 0.3 (partialing out the covariates), the required sample size is c. 73. Finally, hypothesis 4 deals with time-varying predictors in a multilevel model. Sample sizes required for this hypothesis were 109 for sC, 124 for sAA, and 109 for HF-HRV. Regarding missing values, the percentage of missing values in each variable involved in any of the analyses performed varied between 0 and 2.6, except for HF-HRV with a value of 11.4. Little's multivariate test for missing completely at random (MCAR) pattern did not reject the null hypothesis of MCAR

($\chi^2$ = 65.5, df = 67, p = 0.53). When using multilevel models, the missing at random (MAR) pattern is expected to lead to unbiased estimates.

## 3. Results

The proportion of participants who withdrew from participation before the onset of treatment did not differ between women with mental disorders (18%) and healthy controls (21%, $\chi^2$ = 0.24, p = 0.625). However, comparatively more participants dropped out between pre- and posttreatment in the clinical sample (15%) than in healthy controls (1%; $\chi^2$ = 12.80, p < .001). Participants' age at pretreatment did not differ between those who dropped out between pre- and posttreatment (mean = 23.0, SD = 4.2) and those who did not (mean = 22.8, SD = 3.9; t(31)=0.24, p = 0.809).

Descriptives by group and time point of all three outcomes of physiological stress parameters and of all covariates plus BMI and age are shown in Table 1.

### 3.1. Hypothesis 1 (pre-treatment comparisons between specific combinations of groups)

Mean values for sC were higher in the AN group compared to the combined BN/mixed/HC group (a priori contrast, b = 0.259, 95% CI=[0.075, 0.442], t(253)=2.78, p = .006, $p_{adj}$ = .009, d = 0.43). In contrast, mean values for sAA did not differ between the combined BN/mixed group and the combined AN/HC group (a priori contrast, b = 0.111, 95% CI=[–0.154,0.377], t(254)=0.83, p = .409, $p_{adj}$ = .409, d = 0.12). Finally, mean HF-HRV was highest in the combined AN/ BN group and lowest in the mixed group, with intermediate mean values for HC (a priori linear contrast, b = 0.739, 95% CI=[0.302, 1.175], t(224)=3.33, p = .001, $p_{adj}$ = .003, d = 0.617).

When disregarding healthy participants and controlling for the two additional covariates patient type, and comorbidity, mean values for sC were higher in the AN group compared to the combined BN/mixed group (a priori contrast, b = 0.330, 95% CI=[0.115, 0.546], t(153)=3.03, p = .003, $p_{adj}$ = .005, d = 0.55). Mean values for sAA did not differ between the

**Table 1. Descriptives of physiological measures across different subject groups and two time points.**

| | HC | AN | BN | Mixed |
|---|---|---|---|---|
| | mean (SD) N | mean (SD) N | mean (SD) N | mean (SD) N |
| Age | 21.5 (2.2) 101 | 22.4 (4.2) 58 | 23.1 (4.1) 54 | 25.3 (4.8) 59 |
| BMI | 22.1 (2.7) 98 | 17.2 (1.6) 56 | 22.7 (2.5) 54 | 24.3 (6.5) 58 |
| physical activity | 1.31 (0.53) 101 | 1.41 (0.79) 55 | 1.55 (0.67) 54 | 1.05 (0.62) 57 |
| smoking habit | 0.35 (0.66) 97 | 0.76 (1.13) 56 | 1.05 (1.31) 53 | 1.25 (1.31) 59 |
| medication intake | 6 (5.9%) | 17 (29%) | 23 (43%) | 42 (71%) |
| use of contraceptives | 56 (55%) | 27 (47%) | 32 (59%) | 30 (51%) |
| comorbidity | – | 28 (48%) | 29 (54%) | 42 (71%) |
| patient type inpatient outpatient | – | 32 (55%) | 21 (39%) | 45 (76%) |
| | | 26 (45%) | 33 (61%) | 14 (24%) |
| number of therapy sessions | – | 28.2 (48.1) | 13.6 (19.9) | 15.3 (16.3) |
| | | 58 | 54 | 59 |
| sC (pre-treatment) | 1.97 (0.57) 101 | 2.14 (0.59) 58 | 1.86 (0.68) 53 | 1.70 (0.57 59 |
| sC (post-treatment) | 1.66 (0.63) 100 | 2.01 (0.58) 52 | 1.81 (0.64) 47 | 1.74 (0.63) 46 |
| sAA (pre-treatment) | 4.05 (0.94) 101 | 4.60 (0.81) 58 | 4.34 (0.96) 54 | 4.51 (0.90) 59 |
| sAA (post-treatment) | 3.96 (1.00) 100 | 4.39 (0.83) 52 | 4.23 (1.11) 47 | 4.46 (0.83) 46 |
| HF-HRV (pre-treatment) | 6.78 (1.08) 85 | 6.89 (1.39) 55 | 6.48 (1.11) 50 | 5.80 (1.27) 51 |
| HF-HRV (post-treatment) | 6.65 (1.17) 84 | 6.25 (1.18) 49 | 6.00 (1.37) 44 | 6.03 (1.33) 43 |

Notes: AN, anorexia nervosa; BN, bulimia nervosa; HC, healthy group; sC, ln(Cortisol); sAA, ln(Amylase); HF-HRV, ln(heart rate variability).

combined BN/mixed group and the AN group (a priori contrast, b = 0.171, 95% CI=[–0.492,0.151], t(154)=1.05, p = .296, $p_{adj}$ = .296, d = 0.191). Finally, mean values of HF-HRV were higher in the combined AN/ BN group than in the mixed group (b = 0.837, 95% CI=[–0.366, 1.308], t(140)=3.51, p < .001, $p_{adj}$ = .002, d = 0.68). Regarding the impact of covariates, mean values for sC and HF-HRV were lower in case of medication intake when comparing HC and clinical groups. In addition, HF-HRV mean values were also lower when comparing within clinical groups, only in the case of medication intake.. Results concerning covariates are only reported if noteworthy, i.e., if the p-value of the association with the respective outcome was smaller than .01.

### 3.2. Hypothesis 2 (pre-post change in clinical groups relative to HC)

Contrary to our expectation, sC values decreased less in the clinical groups compared to HC (interaction "pre_post x Clinical_vs_HC", b = 0.276, 95% CI=[0.097, 0.455], t(236)=2.99, p = .004, $p_{adj}$ = .008, d = 0.63). sC values decreased from 1.94 to 1.89 in the clinical groups between pre- and posttreatment, and from 1.92 to 1.60 in HC (waiting period for HC). For sAA, the decrease between pre- and posttreatment was comparable between the clinical groups and HC (interaction "pre_post x Clinical_vs_HC", b = –0.079, 95% CI=[–0.302, 0.144], t(234)=0.691, p = .490, $p_{adj}$ = 0.590, d = 0.15). sAA values decreased from 4.49 to 4.33 in the clinical groups between pre- and posttreatment, and from 4.04 to 3.97 in HC. For HF-HRV as expected, the decrease between pre- and posttreatment was more pronounced in AN/BN than after waiting in HC (interaction "pre_post x AN/BN_vs_HC", b = –0.637, 95% CI=[–0.230, –1.044], t(203)=3.03, p = .003, $p_{adj}$ = .008, d = 0.73), but there was no difference in pre-post measures between the mixed group and HC (interaction "pre_post x mixed_vs_HC", b = 0.153, 95% CI=[–0.395, 0.703], t(206)=0.540, p = .590, $p_{adj}$ = .590, d = 0.18). HF-HRV decreased from 6.76 to 6.12 in AN/BN, increased from 6.07 to 6.22 in the mixed group, and remained at 6.58 in HC, between pre- and posttreatment.

   Disregarding healthy participants and controlling for the three additional covariates patient type, number of therapy sessions, and comorbidity, the sC decrease in the combined AN/BN group was comparable to that in the mixed group (interaction "pre_post x AN/BN_vs_mixed", b = 0.038, 95% CI=[–0.259, 0.183], t(136)=0.328, p = .743, $p_{adj}$ = 1.0, d = 0.09). Mean sC values for pre- and posttreatment were 1.98 and 1.91 in the combined AN/BN group, and 1.83 and 1.80 in the mixed group, respectively. For sAA, the decrease between pre- and posttreatment was again comparable between the combined AN/BN group and the mixed group (interaction "pre_post x AN/BN_vs_mixed", b = –0.152, 95% CI=[–0.406, 0.102], t(134)=1.144, p = .255, $p_{adj}$ = .383, d = 0.33). Mean sAA values for pre- and posttreatment were 4.46 and 4.28 in the combined AN/BN group, and 4.53 and 4.50 in the mixed group, respectively. Finally, for HF-HRV, the decrease between pre- and posttreatment was higher in the combined AN/BN group than in the mixed group (interaction "pre_post x AN/BN_vs_mixed", b = –0.843, 95% CI=[–1.424, –0.266], t(127)=2.776, p = .006, $p_{adj}$ = .018, d = 0.82). Mean HF-HRV for pre- and posttreatment were 6.66 and 6.07 in AN/BN, and 5.85 and 6.11 in the mixed group, respectively. Regarding the impact of covariates on the change in outcomes between pre- and posttreatment, none were of importance for any of the three outcomes sC, sAA, and HF-HRV.

### 3.3. Hypothesis 3 (association between BMI and physiological stress parameters for specific combinations of groups)

For sC the correlation with BMI was r = 0.06 (t(49)=0.44, p = .665, $p_{adj}$ = .868) in the AN group and r = –0.02 (t(106)=0.21, p = .837, $p_{adj}$ = 1.0) in the combined BN/mixed group. For sAA, the correlation between BMI and was r = 0.02 (t(158)=0.28, p = .781, $p_{adj}$ = 1.0) in the clinical group (AN, BN, and mixed combined). Finally, for HF-HRV, the correlation with BMI was r = –0.07 (t(46)=0.46, p = .646, $p_{adj}$ = 1.0) in the AN group and r = –0.25 (t(96)=2.58, p = .012, $p_{adj}$ = .060) in the combined BN/ mixed group. Regarding the impact of covariates, none of them were highly associated with any of the outcomes sC, sAA, and HF-HRV at baseline, except that sC values were higher in outpatients than in inpatients in the combined BN/mixed groups.

### 3.4. Hypothesis 4 (BMI is temporally associated with physiological stress parameters in AN)

Neither sC (b=−1.249, 95% CI=[−3.250, 0.753], t(45)=1.233, p=.224, $p_{adj}$=.336, $R^2_{\beta*}$=0.010), nor sAA (b=0.639, 95% CI=[−2.470, 1.193], t(45)=0.690, p=.494, $p_{adj}$=.494, $R^2_{\beta*}$<0.002), nor HF-HRV (b=−4.264, 95% CI=[−9.109, 0.543], t(41)=1.753, p=.087, $p_{adj}$=.261, $R^2_{\beta*}$=0.026) were associated with BMI over time in the AN group.

A noteworthy impact of covariates was only observed for the HF-HRV, which was associated with both the use of medication and of contraceptives. Thus HF-HRV was lower for medication intake than if no medication was taken and higher in patients using contraceptives than not using contraceptives.

## 4. Discussion

This study aimed to compare stress physiology at rest before treatment between women with ED, mixed mental disorders (mostly depression and anxiety), and women without any mental disorders. In addition, we examined the variability of physiological markers of mental health after an evidence-based treatment in different clinics in Switzerland and Germany.

### 4.1. Pre-treatment comparisons between specific combinations of groups

As expected, and in line with the findings of the [10] study, we found higher sC levels at rest in AN compared to the combined group of women with BN, mixed mental disorders, and healthy young women before treatment or the corresponding waiting period. Our findings contrast with the results of [54], who did not describe any sC differences within a sample including AN and BN, (n=27). As these authors did not describe the resting period, the comparison with these findings is limited. Nevertheless, similar sC levels at rest within participants with depression and healthy controls [11,12] or with anxiety [13] have previously been reported and therefore are in line with our expectations.

Our findings did not confirm the expected elevated sAA levels (representing ANS activity) in BN or mixed mental disorders, compared to the group with AN or the healthy group, which is likely due to the high sAA levels in our AN group (Table 1). Therefore, our findings do not support previous studies where sAA levels were enhanced solely in BN, but not in AN in comparison to healthy controls [10] and compared to patients with depression or anxiety [12,33]. However, as resting sAA was not assessed repeatedly in these studies and the sample size, especially in the Monteleone et al. study, was small, it remains difficult to compare our findings with the literature.

Basal vagal tone differed among the four groups, and as hypothesized, basal vagal tone was highest in women with AN or BN, slightly lower in the healthy control group, and lowest in women with mixed mental disorders. These findings support the literature of elevated basal vagal tone in ED, which is probably caused by different ED symptoms, including food restriction and weight control behavior, but also by binge eating behavior, resulting in a desensitization of the autonomic nervous system and elevated vagal tone at rest [17]. Further, our findings confirm the evidence that, in contrast to ED, lower basal vagal tone is expected in affective psychopathologies such as depression and anxiety [25–27] as represented in the mixed group of this study. These levels of basal vagal tone in ED might be the result of ED-related behaviors (food restriction, repeated binge and compensatory behavior) causing an overactivity of the vagal tone in ED patients and of chronic stress levels related to the chronic activation of the fear-defense system in mixed disorders (e.g., [16–19]).

### 4.2. Pre-post change in selected clinical groups relative to HC

Contrary to our expectations, sC levels decreased less in the clinical groups (AN, BN and mixed mental disorders) during therapy than during the waiting period in healthy women, whereas, as expected, the decrease of sAA was comparable between all clinical groups and the healthy sample. These findings might point to the long-lasting effect of ED pathologies on the physiological stress system even after weight normalization [55], which are probably not reversed after a treatment duration of three months. Such treatment-resistant HPA activation patterns have previously been reported by Het and colleagues [37]. They investigated a small sample of 13 young women with ED in comparison with a

non-clinical sample and found at discharge from an in-patient treatment (at an average of 8.47 weeks SEM ± 1.8 (range: 2–58 weeks) after the first assessment) unchanged cortisol responses to a lab-based stress task. In our larger sample of 99 women with ED (52 AN, 47 BN), post-assessment time point was not at discharge of treatment, but defined as a 3-month treatment period, and even after this period, we found less decrease of sC in the clinical samples than in the healthy one. This result might be explained by the few full and partial remissions as they were only rarely achieved in the ED groups, and in the mixed mental control group, the effects of treatment were moderate at best [47]. Therefore, we assume that our findings can be interpreted as ongoing alterations of the psychophysiological stress regulation systems. To which extent this unchanged level of sC might represent a typical pattern of ED physiology or a premorbid vulnerability of ED [37] that remains after treatment resulting in a lack of stress regulation has to be further investigated [37,42]. Such premorbid physiological vulnerability has also been discussed in relation to early traumatic life experiences, e.g., maltreatment [8,56,57], which leads to a heightened sensitivity to subjectively salient stimuli and increased anxious arousal, impacting the stress system and resulting in limited flexibility to adapt to challenging conditions or even blunted stress responses [56]. In addition, such experiences have been associated with heightened cytokine levels, which seem to be linked with lower effects of psychotherapeutic interventions [58]. Besides this, dysregulated cortisol levels are not isolated hormonal changes in ED but are part of a complex hormonal network system that might decelerate HPA normalization during treatment [59]. Therefore, sC changes can probably only be expected within an extended observation window [37] and not after a 3-month treatment.

Nevertheless, as expected, basal vagal tone decreased more between pre- and posttreatment in the AN/BN group than during waiting in healthy controls. The AN/BN group changes might indicate the beginning of a normalization process due to less food restriction and improved weight control, known to cause a desensitization of the physiological system and finally heightened vagal tone, even though full remission of ED-symptoms only rarely occurred in our study [47]. Similar changes have been found in a study by Het and colleagues [37]. Vagal tone normalized during treatment and existing differences between ED and healthy volunteers before in-patient treatment did not exist anymore at discharge [37]. In contrast, data on only BN patients according to a recently published umbrella meta-analysis did not indicate such a normalization after treatment [60]. In our study HRV values of clinical groups changed towards those of HC and we therefore assume that this corresponds with an improvement of parasympathetic activation either as a counter-regulation to still existing high levels of sympathetic activation or due to a reduction of stress causing an autonomic imbalance which might be restored in parallel to ED symptom improvement [37,60]. It is evident, that the mechanisms underlying vagal alterations are multifactorial [60,61] and therefore their understanding still needs further research. However, there is data that brain regions such as the prefrontal cortex and amygdala related to HRV functioning (including basal vagal tone) are also critical in regulating emotions and are positively influenced by psychotherapy contributing to symptom change and probably to a higher autonomic flexibility [60]. Future studies should investigate whether a continuous decrease of ED symptoms after psychotherapy in patients with ED might be related to ongoing improvements in the physiological stress system, as has been observed in depressive and anxiety disorders after CBT [62].

As expected, the change in basal vagal tone in HC before and after the three-month waiting period was comparable to the change in the mixed group during treatment. Similar findings had been reported in a physically healthy sample with depression at post-psychotherapy [63], but not under other treatment conditions such as antidepressants (e.g., [28]). In addition, previous studies on anxiety disorders showed consistent improvements of vagal tone (e.g., meta-analysis of [64]), which let us to assume that the biological defense system is not overactivated anymore after treatment.

Altogether, our findings support the distinct recovery abilities of the two physiological systems and underline the autonomic nervous system to be a faster-changing structure, whereas the HPA axis, being part of a complex hormonal system impacted at several levels by ED.

### 4.3. Association between BMI and physiological stress parameters for specific combinations of groups

We did not find any association of sC at rest and BMI in AN or the combined BN and mixed groups, nor did we find a relation of sAA levels or HF-HRV with BMI in the clinical groups. These findings do not confirm what has been found previously in healthy individuals (e.g., [65]). However, BMI clearly being different in AN from BN or mixed group, we did not find an association with levels of basal vagal tone and therefore assume that HRV in this case rather represents the underlying emotion regulation difficulties which are related to higher medial prefrontal cortex and amygdala functional connectivity [66,67] and to less emotional wellbeing [68]. However, physical condition represented by BMI does not show this association with vagal tone in our study and is rather linked to the symptomatology of the different pathologies in our sample (see hypothesis 1).

### 4.4. The impact of BMI change on physiological measures in AN

The changes in physiological parameters during treatment could not be attributed to changes in BMI in AN, which might represent long-lasting physiological dysregulation in AN due to still ongoing reduced caloric intake within the 3 months of treatment and respective low weight gain [47]. Such a lack of association of BMI and HRV in AN has also been reported in patients without weight restoration [69,70], and cardiac morphological remodeling has only been seen in those with weight recovery [70]. Further, even a short intensive re-nutrition and somatic stabilization intervention (mean length: 41 days) has not shown any cortisol changes in AN [71] and underlines the relevance of full recovery weight in terms of vagal recovery.

### 4.5. Limitations

Besides the strengths of this study, such as the sample size of altogether 275 young women with different mental disorders and healthy controls, the rigorous assessment of mental disorders according to a structured clinical interview, the assessment and consideration of confounders in our analyses, and the allowed heterogeneity of the sample, which better allows to generalize results [72,73], there are also important limitations. Treatment plans, even though following evidence-based guidelines in the present study, varied between patients due to the incorporation of data from multiple clinical sites in different locations [41,47]. Drop-out rates between pre and post-treatment or pre and post-assessment, respectively, were higher in our clinical sample than in our healthy control sample, even though attrition rates in the clinical sample were lower than in other studies, where dropout rates varied from 20.2–51% in inpatient and from 29 to 73% in outpatient ED settings [74]. In addition, our findings apply only to women and cannot be generalized to male populations. It further must be considered that the physiological parameters assessed are known to vary depending on the type of stress exposure at or just before assessment (e.g., acute daily stressors or currently high levels due to the lab condition). Therefore, repeated sampling of sC and sAA was used and calculations of mean levels were considered to better define resting levels, and the assessment period (only between 2 pm and 4.30 pm [75] was set to be limited. Nevertheless, the measures were taken from women anticipating a laboratory experiment on thin ideals and eating behavior, but physiological stress measures remained low in our sample. Besides this, there might be other potential confounders that impact physiological stress responses, e.g., illness duration, circadian rhythm, which could not be considered in our statistical analyses.

## 5. Conclusion

We investigated stress physiology in women with and without mental disorders and potential changes after treatment and in healthy women after a waiting period. As expected, we found indications that psychopathology in women with ED and mixed mental disorders translates into impairment of the HPA axis and the autonomic functioning in a resting condition that did not fully normalize over a 3-month treatment, although symptoms improved in the same period. Especially in severely underweight individuals, we assume that the short-term improvement may be dependent on weight recovery levels, and

therefore, longer follow-up assessment periods are needed. Based on the results obtained in this study, it can be speculated that remaining dysregulated stress physiology after therapy might be related to prolonged recovery periods of the physiological system, also represented in low remission rates. Such physiological dysregulations are likely to impair the patient's capacity to adequately react to daily challenges after in or outpatient treatment. If replicated, this finding underlines the need for ongoing aftercare support to prevent deterioration over time [76]. Such aftercare interventions could target stress system dysfunctions by training related to coping with psychological stress and by implementing breathing, meditation-oriented or biofeedback methods [77,78]. Nevertheless, a focus should also be put on improved aspects of emotion regulation, like alexithymia, as the capability to identify and name emotions or impulsivity is linked to better interoception in people with ED [56,79]. In addition, there is preliminary evidence that training gastric interoception might be linked to ED pathology improvement [80]. Future research would further profit from a harmonization of assessment methodology across studies and a merging of data in shared databases to allow for clearer comparisons of findings [40].

## Acknowledgments

We would like to thank all the participants who contributed data. We also thank all the clinical centers, the students, and the research team for their valuable contributions.

## Author contributions

**Conceptualization:** Simone Munsch, Marius Rubo, Nadine Messerli-Bürgy.

**Data curation:** Simone Munsch, Nadine Messerli-Bürgy, Andrea H. Meyer, Dirk Adolph.

**Formal analysis:** Simone Munsch, Marius Rubo, Andrea H. Meyer.

**Funding acquisition:** Simone Munsch, Silvia Schneider.

**Investigation:** Kathrin Schopf, Nadine Humbel, Felicitas Forrer.

**Methodology:** Simone Munsch, Andrea H. Meyer.

**Project administration:** Simone Munsch, Nadine Messerli-Bürgy.

**Resources:** Simone Munsch.

**Supervision:** Simone Munsch, Nadine Messerli-Bürgy, Kathrin Schopf.

**Validation:** Andrea H. Meyer.

**Writing – original draft:** Simone Munsch, Nadine Messerli-Bürgy.

**Writing – review & editing:** Simone Munsch, Marius Rubo, Nadine Messerli-Bürgy, Andrea H. Meyer, Kathrin Schopf, Nadine Humbel, Felicitas Forrer, Dirk Adolph, Silvia Schneider, Jürgen Margraf.

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
