## [Decision Letter · Decision Letter 0]

18 May 2025

PONE-D-24-58138Resting vagal tone, alpha amylase and cortisol levels in Women with Eating Disorders before and after psychotherapyPLOS ONE

Dear Dr. Munsch,

Thank you for submitting your manuscript to PLOS ONE. After careful consideration, we feel that it has merit but does not fully meet PLOS ONE’s publication criteria as it currently stands. Therefore, we invite you to submit a revised version of the manuscript that addresses the points raised during the review process.

**ACADEMIC EDITOR:**

Thank you for submitting your manuscript to PLOS ONE. The study addresses a highly relevant topic, particularly the physiological impact of psychotherapy in women with eating disorders—an area of growing clinical and theoretical interest. The longitudinal design, the inclusion of multiple stress-related biomarkers, and the multicenter approach are all notable strengths.While the study is ambitious and methodologically solid in many respects, there are conceptual and analytical issues that need to be addressed to strengthen its scientific contribution.Most importantly, the results—although clearly reported—are modest in terms of post-treatment physiological changes. The interpretation of these findings, particularly regarding cortisol and alpha-amylase, would benefit from further critical reflection. The discussion should explore more thoroughly what these null or limited effects imply for the clinical utility of such biomarkers. Likewise, the observed decrease in vagal tone in AN/BN groups is presented as a sign of normalization, but this interpretation requires more cautious framing and stronger theoretical grounding.The heterogeneity of the therapeutic interventions also warrants further attention. Although the manuscript acknowledges variability in setting, duration, and type of therapy, it does not sufficiently consider how these differences may have influenced the results. Addressing this issue in the interpretation—if not in the analyses—would help clarify the study’s conclusions.Furthermore, although sample size calculations are reported, high attrition rates may have affected the statistical power of longitudinal comparisons. A clearer discussion of dropout and how missing data were handled is needed.Finally, several sections of the manuscript—particularly the introduction and discussion—would benefit from more concise and accessible writing to improve overall clarity.The authors are encouraged to revise the manuscript accordingly and to carefully consider the comments provided by the reviewers. The core idea of the study is valuable, and with these improvements, the manuscript could make a meaningful contribution to the field.

We look forward to receiving your revised manuscript.

Kind regards,

Juan Luis Castillo-Navarrete, Ph.D.

Academic Editor

PLOS ONE

2. Please update your submission to use the PLOS LaTeX template. The template and more information on our requirements for LaTeX submissions can be found at http://journals.plos.org/plosone/s/latex .

 [Swiss National Science Foundation, SNSF TSF: 100014L_149416 / 2

German Research Foundation, DFG: SCHN 415 /41

Swiss Anorexia Nervosa Foundation, SANS: 22 12

Research Found University of Fribourg: 419]. 

5. We note that you have indicated that there are restrictions to data sharing for this study. For studies involving human research participant data or other sensitive data, we encourage authors to share de-identified or anonymized data. However, when data cannot be publicly shared for ethical reasons, we allow authors to make their data sets available upon request. For information on unacceptable data access restrictions, please see http://journals.plos.org/plosone/s/data-availability#loc-unacceptable-data-access-restrictions.

Reviewers' comments:

Reviewer's Responses to Questions

**Comments to the Author**

1. Is the manuscript technically sound, and do the data support the conclusions?

Reviewer #1: Yes

Reviewer #2: Partly

2. Has the statistical analysis been performed appropriately and rigorously? 

Reviewer #1: Yes

Reviewer #2: Yes

3. Have the authors made all data underlying the findings in their manuscript fully available?

Reviewer #1: Yes

Reviewer #2: Yes

4. Is the manuscript presented in an intelligible fashion and written in standard English?

Reviewer #1: Yes

Reviewer #2: Yes

5. Review Comments to the Author

Reviewer #1: The paper is well-written and methodologically clear. There is no specific concerns about this manuscript. However, I think it might be improved considering specific aspects:

- The reduction in HF-HRV in AN/BN post-treatment is assumed to indicate normalization, but it is unclear whether this reflects improved autonomic flexibility or simply reduced overactivity.

- Cortisol levels (sC) did not significantly decrease, despite symptom improvement—potential explanations (e.g., chronic stress, metabolic adaptation, HPA-axis dysregulation) should be discussed.

- Please, consider as possible element that should be taken into serious consideration the effects of possible traumatic events that might characterized specific ED people (see https://doi.org/10.1002/erv.2896)

- The study shows higher vagal tone in AN/BN and lower tone in anxiety/depression, but the mechanisms behind these differences are not fully explained.

- Does vagal overactivity in EDs result from chronic malnutrition, metabolic changes, or an adaptive parasympathetic dominance?

- BMI changes were not strongly correlated with stress physiology changes, which contradicts expectations. If BMI alone does not drive physiological normalization, what other factors (e.g., illness duration, stress exposure, psychological distress) play a role?

- The study highlights the need for aftercare, but lacks specific recommendations for improving physiological recovery in EDs.

- Could HRV biofeedback, vagal nerve stimulation, or stress regulation strategies help restore autonomic balance?

- Have you considered the difficulties that people with ED might have in the interoceptive awareness (see https://doi.org/10.1007/s40519-022-01394-7)?

- Some sections are overly technical and dense, requiring clearer phrasing.

- Tables and figures should be formatted consistently for better readability.

Reviewer #2: Dear Authors,

First, I would like to congratulate you on the development of an original and relevant study that contributes to the understanding of stress physiology in women with eating disorders (EDs) and mixed disorders. Below, I detail substantive and specific observations that could further strengthen the manuscript:

- Heterogeneity of treatments (main interpretive limitation): Although treatments followed national clinical guidelines, their duration, modality (inpatient/ambulatory), and number of sessions varied widely. This heterogeneity is recognized as a covariate, but remains a factor limiting causal attribution between treatment and physiological changes. I recommend further analysis of how this variability may have differentially influenced biomarkers. In addition to considering exploratory sub-analyses by treatment modality (e.g., outpatient vs. inpatient).

- Salivary alpha amylase (sAA): expected elevation of sAA in BN and mixed group, but data did not confirm this. could the time of collection (late) or experimental design have influenced this? Discuss how the diurnal sAA curve or anticipatory response to the experimental context might have flattened the expected differences. It may be more informative to use it as an index of acute response to the experimental stressor, rather than measuring basal levels under poorly standardized conditions.

- Salivary cortisol (sC): In the discussion, consider the influence of the circadian rhythm of cortisol (collection between 14:00-16:30 limits interpretation due to the descending phase of the diurnal curve). In addition, it should be mentioned as a limitation that cortisol on awakening (CAR) was not considered as a more robust marker of HPA axis activation in these clinical pictures.

- Underutilized clinical and psychopathological variables: We did not include an analysis by duration of EDs or specific comorbidities, which could modulate the physiological response. If these data are available, it is recommended that they be incorporated into the analysis.

- Absence of associations between BMI and biomarkers: The expected correlations between BMI and vagal tone were not observed in AN. This could reflect limitations of HRV as an index of emotional regulation in severely malnourished individuals. I recommend considering whether physiological dysregulation in AN may be less sensitive to short-term weight variations. Also, discuss whether duration of disorder or chronicity (used as a covariate) might have influenced the physiological response.

- Minor comments: In methodology clarify the use of repeated measures: was the mean of the two saliva samples used for sC and sAA?

In conclusion, the manuscript has potential, but needs some adjustments and clarifications to reach an even higher level of interpretive rigor. In its current form, the findings suggest some dissociation between clinical improvement and physiological modulation, which should be further discussed as an indicator of:

- Persistent dysfunction of stress systems despite symptomatic improvements,

- Or, of methodological limitations in capturing subtle physiological changes over short periods of time.

Sincerely,

Anonymous reviewer

6. PLOS authors have the option to publish the peer review history of their article (what does this mean? ). If published, this will include your full peer review and any attached files.

**Do you want your identity to be public for this peer review?** For information about this choice, including consent withdrawal, please see our Privacy Policy .

Reviewer #1: No

Reviewer #2: No

---

## [Author Response · Author response to Decision Letter 1]

2 Jul 2025

Fribourg, June 30, 2025.

To: Editors of PLOS One

Concerning Resubmission of the Manuscript: “Resting vagal tone, alpha amylase and cortisol levels in Women with Eating Disorders before and after psychotherapy”

Dear Editor,

We would like to thank you for the possibility to resubmit our manuscript “Resting vagal tone, alpha amylase and cortisol levels in Women with Eating Disorders before and after psychotherapy”. The comments of the reviewers were very helpful, and we hope that the answers to the reviewers and the changes in the manuscript are satisfactory. The study was approved by a local ethics committee and conforms to the principles expressed in the Declaration of Helsinki. Written informed consent was obtained from all participants. This paper reports original, unpublished work that is not considered for publication elsewhere. We declare to have no potential conflict of interest, financially or otherwise. The present research was supported by grants from several funding agencies (SNSF 149416 and 170503, DFG SCHN 415/4, SANS 22 12, Research Found University of Fribourg: 419), but the funding agencies were not involved in our study design, data collection and analysis, decision to publish, or preparation of the manuscript.

Please let us know if we can be of any further assistance: simone.munsch@unifr.ch; nadine.messerli-burgy@unil.ch

On behalf of all authors,

Shared first authors:

Dr. Simone Munsch

CH-University of Fribourg

simone.munsch@unifr.ch

Dr. Nadine Messerli

CH-University of Lausanne

nadine.messerli-burgy@unil.ch

Point-by-point reply

ACADEMIC EDITOR:

Editor: Thank you for submitting your manuscript to PLOS ONE. The study addresses a highly relevant topic, particularly the physiological impact of psychotherapy in women with eating disorders—an area of growing clinical and theoretical interest. The longitudinal design, the inclusion of multiple stress-related biomarkers, and the multicenter approach are all notable strengths.

While the study is ambitious and methodologically solid in many respects, there are conceptual and analytical issues that need to be addressed to strengthen its scientific contribution.

Response: We thank the reviewer for this overall positive evaluation and agree that we need to rule out uncertainty regarding the conceptual approach to our research questions and have included most of the suggestions of the editor and the reviewers.

Editor: Most importantly, the results—although clearly reported—are modest in terms of post-treatment physiological changes. The interpretation of these findings, particularly regarding cortisol and alpha-amylase, would benefit from further critical reflection. The discussion should explore more thoroughly what these null or limited effects imply for the clinical utility of such biomarkers. Likewise, the observed decrease in vagal tone in AN/BN groups is presented as a sign of normalization, but this interpretation requires more cautious framing and stronger theoretical grounding.

Response: We agree with the editor's summary. We discuss the meaning of our findings in more detail. It says now on p.23:

Contrary to our expectations, sC levels decreased less in the clinical groups (AN, BN and mixed mental disorders) during therapy than during the waiting period in healthy women, whereas, as expected, the decrease of sAA was comparable between all clinical groups and the healthy sample. These findings might point to the long-lasting effect of ED pathologies on the physiological stress system even after weight normalization (Holsen et al., 2014), which are probably not reversed after a treatment duration of three months. Such treatment-resistant HPA activation patterns have also been reported by Het and colleagues (Het et al., 2020). They investigated a small sample of 13 young women with ED in comparison with a non-clinical sample and found at discharge from an in-patient treatment (at an average of 8.47 weeks SEM ± 1.8 (range: 2–58 weeks) after the first assessment) unchanged cortisol responses to a lab-based stress task. In our larger sample of 99 women with ED (52 AN, 47 BN), post-assessment time point was not at discharge of treatment, but defined as a 3-month treatment period, and even after this period, we found less decrease of sC in the clinical samples than in the healthy one. This result might be explained by the few full and partial remissions as they were only rarely achieved in the ED groups, and in the mixed mental control group, the effects of treatment were moderate at best (Schopf et al., 2023). Therefore, we assume that our findings can be interpreted as ongoing alterations of the psychophysiological stress regulation systems. To which extent this unchanged level of sC might represent a typical pattern of ED physiology or a premorbid vulnerability of ED (Het et al., 2020) that remains after treatment resulting in a lack of stress regulation has to be further investigated (Het et al., 2020; Munsch et al., 2021). Such premorbid physiological vulnerability has also been discussed in relation to early traumatic life experiences, e.g., maltreatment (Lo Sauro and colleagues, 2008; Monteleone et al., 2020; Meneguzzo et al., 2021), which leads to a heightened sensitivity to subjectively salient stimuli and increased anxious arousal, impacting the stress system and resulting in limited flexibility to adapt to challenging conditions or even blunted stress responses (Meneguzzo et al., 2021). In addition, such experiences have been associated with heightened cytokine levels, which seem to be linked with lower effects of psychotherapeutic interventions (Dalton et al., 2018). Besides this, dysregulated cortisol levels are not isolated hormonal changes in ED, but are part of a complex hormonal network system that might decelerate HPA normalization during treatment (Culbert et al., 2016). Therefore, sC changes can probably only be expected within an extended observation window (Het et al., 2020) and not after a 3-month treatment.

In addition, due to the comment of the editor regarding the utility of biomarkers in these clinical groups, we have added further information on page 28, it. Says now:

We investigated stress physiology in women with and without mental disorders and potential changes after treatment and in healthy women after a waiting period. As expected, we found indications that psychopathology in women with ED and mixed mental disorders translates into impairment of the HPA axis and the autonomic functioning in a resting condition that did not fully normalize over a 3-month treatment, although symptoms improved in the same period. Especially in severely underweight individuals, we assume that the short-term improvement may be dependent on weight recovery levels, and therefore, longer follow-up assessment periods are needed. Based on the results obtained in this study, it can be speculated that remaining dysregulated stress physiology after therapy might be related to prolonged recovery periods of the physiological system, also represented in low remission rates. Such physiological dysregulations are likely to impair the patient’s capacity to adequately react to daily challenges after in or outpatient treatment. If replicated, this finding underlines the need for ongoing aftercare support to prevent deterioration over time (Hegedüs et al., 2020). Such aftercare interventions could target stress system dysfunctions by training related to coping with psychological stress and by implementing breathing, meditation-oriented or biofeedback methods (Wendtz & Thayer, 2024) Mathersul et al., 2022). Nevertheless, a focus should also be put on improved aspects of emotion regulation, like alexithymia, as the capability to identify and name emotions or impulsivity is linked to a better interoception in people with ED (Martin et al., 2019; Meneguzzo et al., 2022). In addition, there is preliminary evidence that training gastric interoception might be more closely linked to ED pathology improvement (Tiemann et al., in press). Future research would profit from a harmonization of assessment methodology across studies and a merging of data in shared databases to allow for clearer comparisons of findings (Laufer et al., 2018).

Editor: The heterogeneity of the therapeutic interventions also warrants further attention. Although the manuscript acknowledges variability in setting, duration, and type of therapy, it does not sufficiently consider how these differences may have influenced the results. Addressing this issue in the interpretation—if not in the analyses—would help clarify the study’s conclusions.

Response: Based on the comments of the editor and the reviewers, we now provide more detail on the influence of the therapeutic setting on the biological correlates of stress regulation and including comorbidity in the analyses. The information is provided on different pages in the manuscript. We added more information on our reasoning about the study design and limitations in our response to the reviewers, as we decided not to include the illness duration information, due to a high level of missing data. We comment on this point in the discussion section (p. 26). We further did not include covariates linked closely to the core symptoms of the disorders, such as psychological distress, which is an integral part of the psychopathology of our sample. Controlling for medication intake, physical activity, smoking habit, medication intake with effects on the HPA axis, use of contraceptives, patient type, comorbidity, and number of treatment sessions did not change the main findings (p.17-21).

Explanations are first provided in the method section, where it says now on page 14: “The following covariates were included for all analyses: physical activity, smoking habit, medication intake with effects on the HPA axis, and use of contraceptives. In addition, comorbidity (y/n) and patient type (out- or inpatient treatment/healthy) were considered for hypotheses focusing on comparisons exclusively between the clinical groups. Further, we considered the number of therapy sessions as a covariate whenever changes during treatment were at the center of our analyses including only clinical groups as in parts of hypotheses 2 and 4. Results concerning covariates are only reported if noteworthy, i.e., if the p-value of the association with the respective outcome was smaller than .01. Effect sizes are reported for all results.”

For hypothesis 1, p.18: “When disregarding healthy participants and controlling for the two additional covariates patient type, and comorbidity, mean values for sC were higher in the AN group compared to the combined BN/mixed group (a priori contrast, b=0.330, 95% CI=[0.115, 0.546], t(153)=3.03, p=.003, padj=.005, d=0.55). Regarding the impact of covariates, mean values for sC and HF-HRV were lower in case of medication intake when comparing HC and clinical groups. In addition, HF-HRV mean values were also lower when comparing within clinical groups, only in the case of medication intake. In addition, outpatients exhibited higher values for sC than inpatients. All other covariates were never highly associated with any of the three outcomes sC, sAA, and HF-HRV in hypothesis 1.”

For hypothesis 2, p. 19: “Disregarding healthy participants and controlling for the three additional covariates patient type, number of therapy sessions, and comorbidity, the sC decrease in the combined AN/BN group was comparable to that in the mixed group (interaction "pre_post x AN/BN_vs_mixed", b=0.038, 95% CI=[–0.259, 0.183], t(136)=0.328, p=.743, padj=1.0, d=0.09). … Regarding the impact of covariates on the change in outcomes between pre- and posttreatment, none were of importance for any of the three outcomes sC, sAA, and HF-HRV.”

For hypothesis 3, p.20: “Regarding the impact of covariates, none of them were highly associated with any of the outcomes sC, sAA, and HF-HRV at baseline, except that sC values were higher in outpatients than in inpatients in the combined BN/mixed groups.”

For hypothesis 4, p.21: “A noteworthy impact of covariates was only observed for the HF-HRV, which was associated with both the use of medication and of contraceptives. Thus HF-HRV was lower for medication intake than if no medication was taken and higher in patients using contraceptives than not using contraceptives.”

Editor: Furthermore, although sample size calculations are reported, high attrition rates may have affected the statistical power of longitudinal comparisons. A clearer discussion of dropout and how missing data were handled is needed.

Response: We agree with the editor that attrition rates warrant consideration. Nevertheless, we found in our study that an equal number of women with or without mental disorders withdrew from participation before the study started, while 15% of all participating patients, compared to 1% of the women without mental disorders, dropped out from pre to posttreatment. We did not find any differences in age between those who stayed and those who prematurely terminated their participation.

A review by Fassino and colleagues reported clearly higher dropout rates than in our study, mounting up to 51% for inpatients and up to 79% for outpatient settings. But the authors found only a few differences between dropouts and ED patients who terminated their treatment. Only in terms of the tendency to binge purge and to emotionally overreact, dropouts differed from those who remained in the studies (Fassino et al., 2009).

To provide more information in our manuscript and describe our methodological approach to the attrition rates, we added now the following paragraph to the data analysis section on page 15: “Regarding missing values, the percentage of missing values in each variable involved in any of the analyses performed varied between 0 and 2.6, except for HF-HRV with a value of 11.4. Little's multivariate test for missing completely at random (MCAR) pattern did not reject the null hypothesis of MCAR (c2=65.5, df=67, p=0.53). When using multilevel models, the missing at random (MAR) pattern is expected to lead to unbiased estimates.”

Editor: Finally, several sections of the manuscript—particularly the introduction and discussion—would benefit from more concise and accessible writing to improve overall clarity.

Response: We revised the introduction and discussion section according to the reviewers and hope that this has improved the clarity of our writing.

Editor: The authors are encouraged to revise the manuscript accordingly and to carefully consider the comments provided by the reviewers. The core idea of the study is valuable, and with these improvements, the manuscript could make a meaningful contribution to the field.

Response: We thank you for the opportunity to revise the manuscript. We have considered all the comments, integrated information where needed, adapted the text and responded to each comment of the reviewers (see below).

Review Comments to the Author

Reviewer #1:

Reviewer: The paper is well-written and methodologically clear. There is no specific concerns about this manuscript. However, I think it might be improved considering specific aspects:

Response: We thank the reviewer for this feedback and the thoughtful evaluation of our manuscript.

Reviewer: The reduction in HF-HRV in AN/BN post-treatment is assumed to indicate normalization, but it is unclear whether this reflects improved autonomic flexibility or simply reduced overactivity.

Response: We thank the reviewer for this comment. It is indeed difficult to assume whether changes of HF-HRV are the result of a sympathetic reduction induced by a lower stress level or rather a result of a high counterbalancing reaction in chronic stress conditions and therefore an overactivity. As we consider the HC group as characterized by high levels of mental health and therefore high functioning of the autonomic nervous system, we conclude that the approaching HF-HRV levels of the clinical samples towards the HC group levels rather represent a normalisation and therefore an increase of autonomic flexibility. We have now changed one part of the discussion section, and it says now on pages 24-25:

“Nevertheless, as expected, basal vagal tone decreased more between pre- and posttreatment i

---

## [Decision Letter · Decision Letter 1]

18 Jul 2025

Resting vagal tone, alpha amylase and cortisol levels in Women with Eating Disorders before and after psychotherapy

PONE-D-24-58138R1

Dear Dr. Simone Munsch,

We’re pleased to inform you that your manuscript has been judged scientifically suitable for publication and will be formally accepted for publication once it meets all outstanding technical requirements.

Kind regards,

Juan Luis Castillo-Navarrete, Ph.D.

Academic Editor

PLOS ONE

Additional Editor Comments:

The authors have thoughtfully addressed the reviewers' suggestions, resulting in a clearer and more compelling manuscript. This work offers meaningful insight into the physiological effects of psychotherapy in women with eating disorders. Congratulations to the authors for this valuable contribution to the field.

Reviewers' comments:

Reviewer's Responses to Questions

**Comments to the Author**

1. If the authors have adequately addressed your comments raised in a previous round of review and you feel that this manuscript is now acceptable for publication, you may indicate that here to bypass the “Comments to the Author” section, enter your conflict of interest statement in the “Confidential to Editor” section, and submit your "Accept" recommendation.

Reviewer #1: All comments have been addressed

Reviewer #2: All comments have been addressed

2. Is the manuscript technically sound, and do the data support the conclusions?

Reviewer #1: Yes

Reviewer #2: Yes

3. Has the statistical analysis been performed appropriately and rigorously? 

Reviewer #1: Yes

Reviewer #2: Yes

4. Have the authors made all data underlying the findings in their manuscript fully available?

Reviewer #1: Yes

Reviewer #2: No

5. Is the manuscript presented in an intelligible fashion and written in standard English?

Reviewer #1: Yes

Reviewer #2: Yes

6. Review Comments to the Author

Reviewer #1: I recommend acceptance with no further revisions. The authors have made thoughtful and rigorous adjustments to their manuscript, directly addressing each concern raised by the editor and reviewers. The methodological transparency, improved interpretative depth, and enhanced clarity of the text significantly strengthen the contribution of this study to the field of psychophysiological research in eating disorders.

Reviewer #2: Thank you for addressing the comments raised in the previous review round so satisfactorily. You responded clearly and substantively to each issue relating to treatment heterogeneity, biomarker interpretation (e.g. salivary alpha-amylase and cortisol) and contextualising results within the clinical complexity of eating disorders. Recognising the limitations of this context considerably improves the interpretive rigour of the manuscript. The new version makes a valuable contribution to this area of research. It is clearly written, sufficiently robust and meets publication standards. Therefore, I support its acceptance for publication.

7. PLOS authors have the option to publish the peer review history of their article (what does this mean? ). If published, this will include your full peer review and any attached files.

**Do you want your identity to be public for this peer review?** For information about this choice, including consent withdrawal, please see our Privacy Policy .

Reviewer #1: No

Reviewer #2: No

---

## [Editor Report · Acceptance letter]

PONE-D-24-58138R1

PLOS ONE

Dear Dr. Munsch,

I'm pleased to inform you that your manuscript has been deemed suitable for publication in PLOS ONE. Congratulations! Your manuscript is now being handed over to our production team.

Kind regards,

on behalf of

Dr. Juan Luis Castillo-Navarrete

Academic Editor

PLOS ONE